# Modeling of Progressive Scouring of a Pier-on-Bank

Vidya Subhash Chavan [1,*], Shen-En Chen [2], Navanit Sri Shanmugam [1], Wenwu Tang [3], John Diemer [4], Craig Allan [4], Nicole Braxtan [2], Tarini Shukla [1], Tianyang Chen [4] and Zachery Slocum [4]

[1] INES (Infrastructure and Environmental Systems) PhD Program, Department of Civil and Environmental Engineering, University of North Carolina at Charlotte, Charlotte, NC 28223, USA; nshanmug@uncc.edu (N.S.S.); tshukla@uncc.edu (T.S.)

[2] Department of Civil and Environmental Engineering, University of North Carolina at Charlotte, Charlotte, NC 28223, USA; schen12@uncc.edu (S.-E.C.); nbraxtan@uncc.edu (N.B.)

[3] Center for Applied Geographical Information Sciences (CAGIS), Department of Geography and Earth Sciences, University of North Carolina at Charlotte, Charlotte, NC 28223, USA; wtang@uncc.edu

[4] Department of Geography and Earth Sciences, University of North Carolina at Charlotte, Charlotte, NC 28223, USA; jadiemer@uncc.edu (J.D.); cjallan@uncc.edu (C.A.); tchen19@uncc.edu (T.C.); zslocum@uncc.edu (Z.S.)

* Correspondence: vchavan1@uncc.edu

**Abstract:** Scour, caused by swiftly moving water, can remove alluvial sediment and soil, creating holes surrounding a bridge component and compromising the integrity of the bridge structure. Such problems can be equally critical for bridges with piers-on-bank bridges subjected to severe storm and flooding issues. In this paper, the Phillips Road Bridge over Toby Creek (35°18′28.2″ N 80°44′16.6″ W, Charlotte, NC, USA), a pier-on-bank bridge with critical/significant local scour holes and deep riverbank erosion cuts was selected as case study bridge. To investigate the scour effect on the bridge with pier-on-bank performance, the scoured area around a single pier is first quantified using a terrestrial laser and then modeled using nonlinear finite element (FE) analysis, where the local scour is modeled as progressive mass losses using the Element Removal (ER) technique. The FE results are compared to the design loading scenario and the results substantiated that the local scouring could cause large deflection and increased bending moment on the bridge pier.

**Keywords:** local scour; lidar scan; bridge piers; finite element

## 1. Introduction

Scour is a critical condition change for a bridge hydraulic system, especially during storms and high water. Scour, caused by swiftly moving water, can remove alluvial sediment and soil, creating holes surrounding a bridge component and compromising the integrity of a structure [1]. Scour associated with bridge piers usually starts out as local scour(s) and is often associated with acceleration of flow and resulting turbulent vortices. Local scour typically starts as a scour hole surrounding the bridge pier [2]. If not addressed, local scours can worsen and result in enlarged mass losses surrounding the bridge supports. The danger of bridge scour failures lies in the fact that they can occur without prior warning. Thus, there is a need for an effective monitoring strategy to identify scour problems surrounding a bridge structure.

The geography of North Carolina falls in three divisions—The western Appalachian Mountains, the central Piedmont Plateau and the eastern Coastal Plain. This diverse landscape resulted in a significant number of bridges with the unique design of piers-on-bank. Figure 1 shows several examples of piers-on-bank bridges from the central Piedmont area, North Carolina.

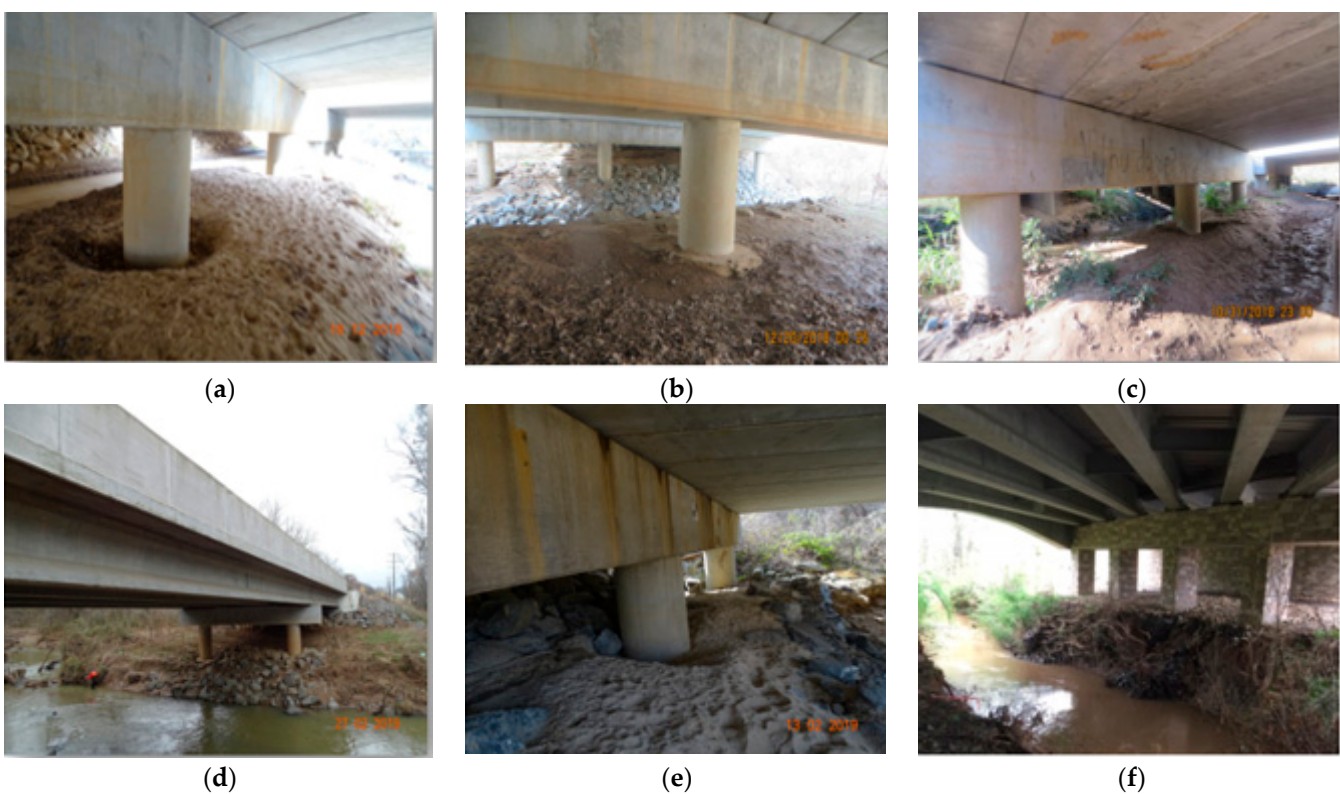

**Figure 1.** Examples of bridges with piers-on-bank and experiencing scour (**a**) Rocky River; (**b**) Mallard Creek; (**c**) Pharr mill; (**d**) Blackwood Creek; (**e**) Rankin Lake; (**f**) Toby Creek.

Most of the piers-on-bank bridges span over high banks surrounding small creeks with low volume flow. However, during severe storms and high floods, the overflowing and rapid river flow would result in scouring at the bridge piers. Contrasting with piers in the river basin, the piers-on-bank experiences scour only during high-waters, and dry debris may accumulate in the scour holes until the next high-water. Figure 2 shows the envisioned mechanisms of local scour surrounding a single bridge pier, where the soil/sediment mass may be removed due to high water velocities and increased turbulence (which are functions of the hydrodynamic characteristics of the river flow), and the competence of the geomaterial surrounding the bridge pier to resist the scouring process [3]. The wet-dry cycle may present a different damaging effect to the piers and currently no studies have been identified for such a bridge. Chavan et al. [4] studied the scouring effects on multiple piers and showed evidence of local and combined scours on piers-on-bank.

To minimize the risk of bridge failure, Departments of Transportation (DOTs) are interested in comprehensive and accurate methods to assess existing bridge conditions so that immediate actions may be taken and help develop remediation plans to minimize risks to safety and finances. However, current efforts to quantify scour effects on bridges during bridge inspections are limited. According to the National Bridge Inspection Standard (NBIS), bridges over 6 m in length must be inspected and rated every other year. Based on the Standard, NBI item 60 for substructure condition is rated on a scale of 0 to 9 where 0 is a failed state beyond corrective action and 9 means excellent condition [5]. Furthermore, NBI Item 113 for scour-critical bridges is rated from 0 being scour critical, to 9 where the bridge foundation is well above floodwater elevations. Thus, the current bridge scour assessment is insufficient to capture the true state of scour and the potential dangers to the bridge.

Current NBI does not differentiate pier locations and the only scour quantifier used in Item 113 is scour depth and not the extent of the scour. Extensive research has been conducted to assess the scour conditions based on maximum observed scour depth in the past, including prediction of future scour depth based on laboratory experiments and theoretical methods. For example, Bridge Scour Assessment method (BSA-1) has been

widely used as an observational method based on measured scour data and observed or estimated flow parameters for estimating future scour depth instead of site-specific erosion testing [6]. Kishore at al. [7] performed laboratory tests to study the effects of scouring on laterally loaded piles. Yang et al. [8] considered fluid-soil interaction in scour stability evaluation of bridge piers under different scour depths and flow velocity conditions. Beg [9] conducted extensive experimental studies of local scouring around two piers placed in the transverse direction to the flow. Many studies have shown that the effect of scour hole dimensions on a pile's lateral response is more critical than other responses, and scour depth alone is not sufficient to quantify scour damage level [10]. However, most of the studies conducted were based on either calculated scour depth or merely considered scour depth.

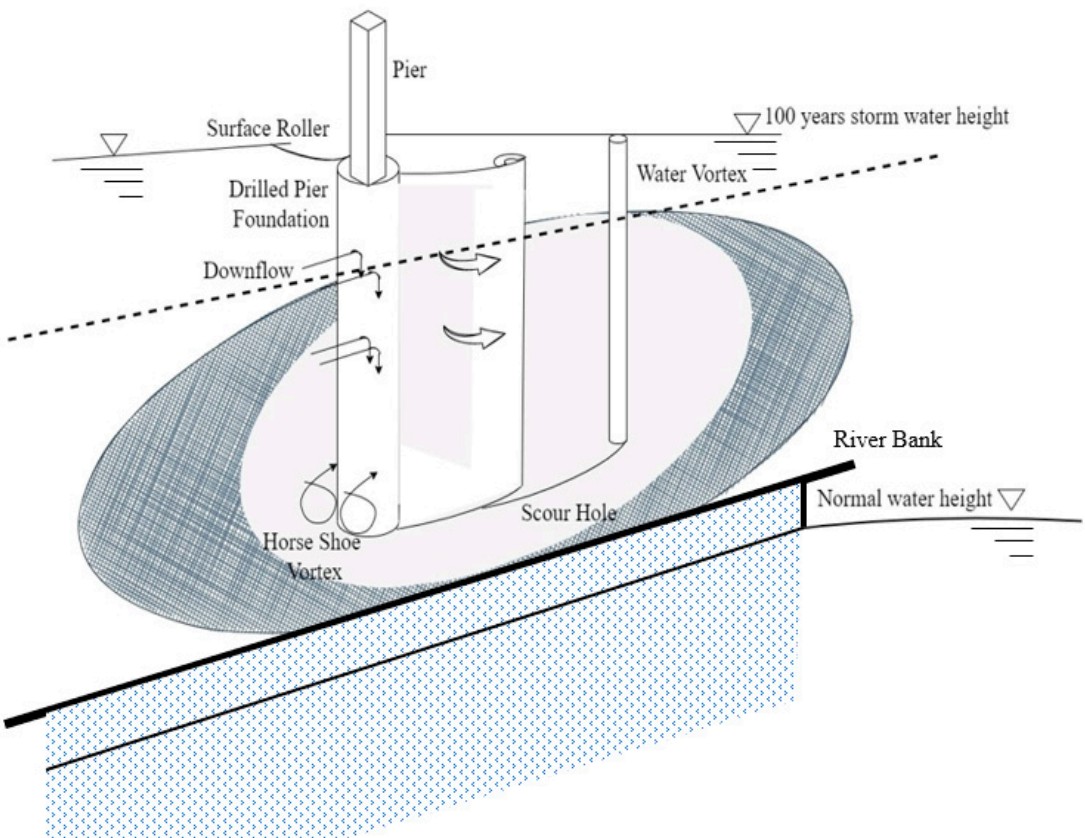

**Figure 2.** Scour mechanism around a single pier.

To study the effects of extreme scour conditions on the performances of bridge foundation and bridge superstructure, scour effects on buckling capacity of a bridge pier have been studied by Avent et al. [11]. McConnell et al. [12] investigated scour effects on the pushover behavior of a bridge. Lin et al. [13] proposed an integrated analysis technique to study the performance of pile-supported bridges under scoured conditions. Finally, Khandel et al. [14] developed a deep learning based integrated neural network for the assessment of different flood hazard intensities to simulate structural behavior of a bridge foundation under scour condition.

Scouring effect is critically dependent on the soil types and several studies have been focused on either cohesive or non-cohesive unconsolidated geologic materials. Lin et al. [15] considered stress history effects of sand erosion on the laterally loaded piles. Liang et al. [16] studied the effects of extreme scour on the buckling of bridge piles considering the stress history of soft clay. Ben et al. [17] demonstrated the effect of stress history in evaluating scour effects on lateral behavior of monopiles in soft clay.

Other than field investigations, numerical methods [15] have been used to investigate the changes in structural responses, including shear stresses, bending moments, pile head displacement, and rotation before and after scouring. Lin et al. [2] developed a close-form solution for the estimation of vertical effective stress and pile lateral capacities considering scour hole depth, width, and slope angle. Majumder et al. [18] used lower bound finite element limit analysis to assess the scour impact on under-reamed pile in clay. Their study demonstrated significant reduction in bearing and uplift capacities of under-reamed pile considering stress history of the clay.

This paper reports on a study involving the scour problem for a piers-on-bank bridge —The Phillips Road bridge (Figure 3a,b) on the UNC Charlotte campus. The three-span continuous prestressed concrete girder bridge has a deck that is 50.5 m in length that spans over Toby Creek. The two-lane bridge was completed and opened in March 2016. Two sets of bridge piers supporting the bridge were built on the two banks of Toby Creek [4].

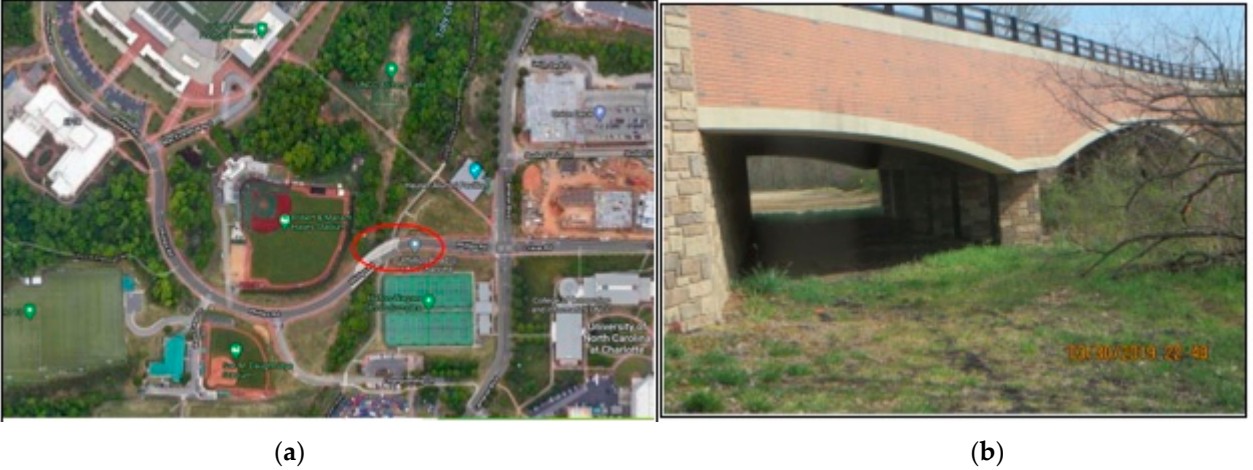

(**a**) (**b**)

**Figure 3.** Location and view of Phillips Road bridge: (**a**) aerial image of the bridge site site (outlined in red), north toward top of aerial photo, and (**b**) downstream view of the western end of the bridge structure, view toward north and parallel to flow direction of Toby Creek (photo credit: S.E. Chen).

The constrained channel size, an increasingly impervious urbanizing catchment in combination with episodic torrential rains have resulted in significant turbulent overflow from river flooding that caused the formation of local scouring surrounding the north side bridge piers. Figure 4 shows one of the bridge piers where two scour mechanisms are occurring concurrently: One involves the localized scour hole formation around the bridge pier, while the other involves lateral bank erosion (contraction scour) along the entire bank face.

To determine the dimensions of the local scour problem, terrestrial LiDAR scans of the bridge piers have been conducted periodically to determine the extent and evolution of the scours. Terrestrial LiDAR has been used for bridge monitoring, including bridge deflection under static loading [19], detection of bridge defects [20], bridge clearance measurements [21], and, more recently, for scour quantification [22]. Terrestrial LiDAR scans can provide high-resolution point cloud data of a bridge hydraulic structure, which can then be used to quantify material losses surrounding a scoured bridge pier. In addition, rapid and repeated laser scans can generate periodic quantification of scours and help define the process of erosion and determine the rate of removal of a streambed or bank material surrounding the bridge foundation.

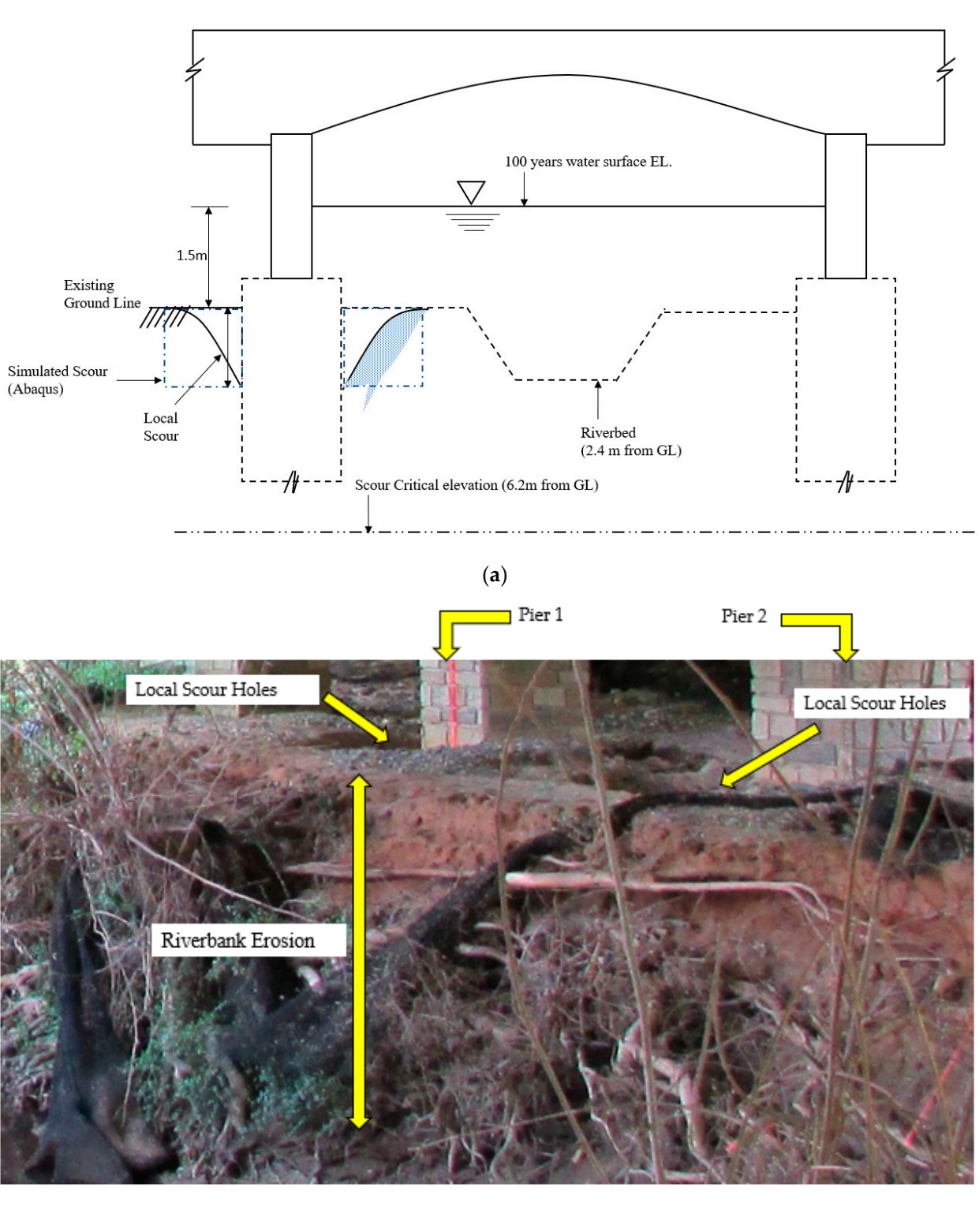

(**a**)

(**b**)

**Figure 4.** Phillips Road bridge scour development: (**a**) Schematics of the local scour formation and simulation (view to north with west end of bridge on the left) and (**b**) local scouring around bridge pier and channel erosion after a torrential rain on west bank of Toby Creek (view to southwest) (photo credit: S.E. Chen).

To study the effects of the scour to the bridge pier, this paper reports on a study of a single bridge pier undergoing active scouring using the finite element method. The scouring extent is first determined using LiDAR scans, which is then idealized as a square scour hole surrounding the bridge pier. A nonlinear Finite Element (FE) method using element removal (ER) technique is then used to simulate the scouring effect. The ER technique quantifies the scour as soil mass losses, which can have an impact on the bridge pier deflection and moment distribution. The loading scenario is determined from the original bridge design reports and is applied and compared with the FE results.

In this paper, the three-dimensional FE method is used first to develop a base model to simulate the substrate-pile interaction of the drilled pier subjected to combined loading. This model is then verified and developed further to simulate and study different scour scenarios for the case study bridge.

### 1.1. Numerical Method/Scour Modeling

A review of bridge scour analysis methods shows that the most widely used technique is the p-y method of analysis of laterally loaded piles [15], which uses p-y curves developed from full-scale test results. The method assumes a beam on a Winkler-foundation model and uses p-y, t-z, and q-z curves to characterize the pile's lateral, axial, and end bearing responses. Using the p-y method, the geologic substrate is typically considered as a series of nonlinear springs spaced at regular intervals along the pile length. The p-y curves used in commercial software are mostly derived from field experiments conducted on various soil conditions, although user-defined p-y curves can also be used to model the soil responses. However, the lateral soil resistance based on the p-y method cannot consider interactions between individual soil elements. Moreover, the shearing forces at the interface between the pile and surrounding substrate are also neglected, as is the case of the solutions proposed by Poulos [23]. Also, as the p-y method cannot consider scour-hole dimensions/extent of the scour hole, the scour width effects are approximated based on the estimation of effective soil stress around the pile [13].

The FE method is an effective tool to study the soil-pile interaction involving the scouring problem. The FE method provides the capability of considering continuity of the soil mass, appropriate nonlinear material models for both the pile and geologic substrate, defining different boundary conditions and nonlinear interaction effects necessary to model the soil-pile contact problems. Kim et al. [24], Mardfekri et al. [25], Strömblad [26], Salim [27], and Youssouf et al. [28] have employed three-dimensional FE methods to study the effect of soil-pile interaction on laterally loaded piles. Senturk et al. [29] performed three-dimensional finite element push-over analyses of bridge piers considering the nonlinear behavior of reinforced concrete and soil under quasi-static loading. Khodair et al. [30] compared the results obtained from the Finite Difference (FD) method and FE method to study the effect of pile-soil interaction under axial and lateral loads. Finally, Peiris et al. [31] studied pile behavior under seismic excitations using the FE method.

### 1.2. Phillips Road Bridge Study

1.2.1. The Study Site

The Phillips Road bridge (Figure 3a,b) has a clear roadway width of 9.8 m and supports two traffic lanes of 4.9 m width each. The overall width of the bridge deck is 15.5 m. The cast-in-situ concrete slab (514.4 mm uniform thickness) is supported by seven prestressed concrete girders on top of the three bridge spans. The intermediate bents are supported on drilled pier foundations, while the end bent abutments are founded on pile-supported strip footing.

Phillips Road bridge spans over Toby Creek, which is a headwater tributary rises in the Newell community of Charlotte, North Carolina, and drains approximately 13.3 km$^2$ and discharges to the Mallard Creek, a tributary of the Yadkin-PeeDee River system. Toby Creek has an estimated average discharge of 0.17 m$^3$/s and a mean flow velocity of 0.274 m/s [32]. The total stream length is 6.68 km. The width of the creek at the bridge at the low flow

stage is approximately 3.0 m and has a maximum bank full depth of 2.1 m. The constricted river cross-section underneath the bridge, in combination with high flow velocities during episodic runoff events, has resulted in significant bank erosion and has induced localized scour at the piers on both streambanks (Figure 4b). Figure 4a shows the river cross-section below the Phillips Road bridge.

The Phillips Road bridge piers and embankments have undergone many cycles of floods in the recent past. As a result, lidar scans taken over a period of two years have revealed local scour of approximately 1.1 m and 1.5 m diameters near bridge piers on both sides of the river channel. Although the depth of local scour holes observed is not more than 1.5 m, on the opposite sides of the piers (channel side), lateral erosion of up to 3–3.4 m has been observed. The observed loss of embankment soil or riverbank is non-uniform along the two pier bents and could be broadly categorized as contraction scour. Thus, the piers on the northwest side of the case study bridge demonstrated a combination of local and contraction scour. This combined scour problem is worsened due to the accumulation of large quantities of debris in the river channel, which would likely serve to further increase the scour volume in future runoff events.

### 1.2.2. Scour Assessment Using LiDAR Scans

To quantify the extent of the scoured area, a FARO Focus S 350 LiDAR was used. FARO LiDAR uses a mono-dyne laser with a wavelength of 1550 nm. Vegetation and other obstacles covering the target must be removed before the scan. Due to the geometric shape of a scour, a full scan of the scour cannot be made from a single scan. Hence, the LiDAR must be shot from multiple angles while keeping in mind the scanning angles and the height of the laser head. Then, they should be merged or "stitched" together to reveal a complete picture. This LiDAR device has a maximum scanning range of 350 m consisting of millions of cloud points. So, the scan was segmented to capture just the region of interest, which is the scour. Figure 5 shows the point cloud of the scour surrounding the selected pier of the Phillips Road bridge from a single scan. Figure 5b shows the localized scour region/hole scanned using LiDAR. As seen in Figure 5c, the entire scour area cannot be observed due to obstacles (both Figure 5b and Figure 5c are for pier number 1). However, the maximum depth and the diameter can be obtained from this scan. The point cloud data is then used to quantify (surface area and volume) the scour by defining a reference plane. Figure 5a shows a sample reference plane drawn to quantify the scour. A detailed description of the mass loss quantification method from LiDAR scans can be found in Liu et al. [20].

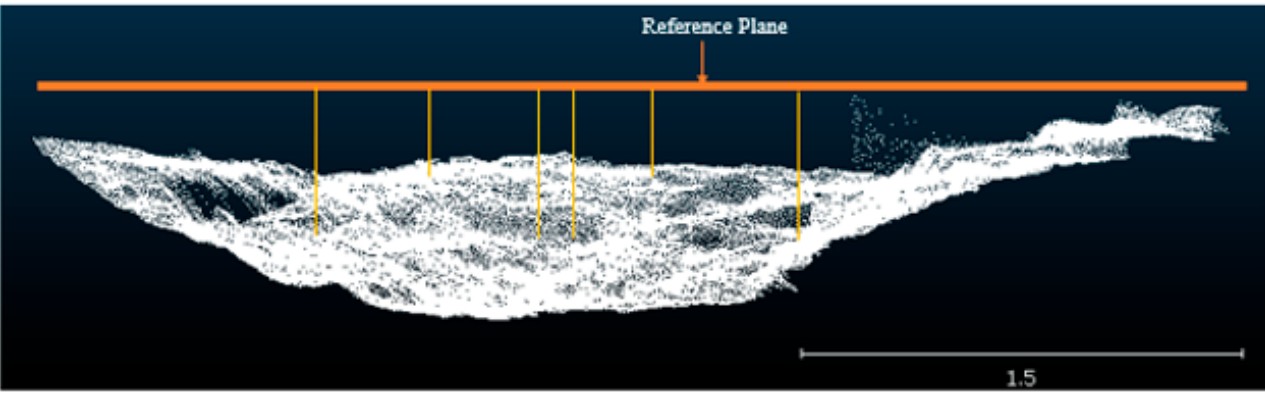

(**a**)

**Figure 5.** *Cont.*

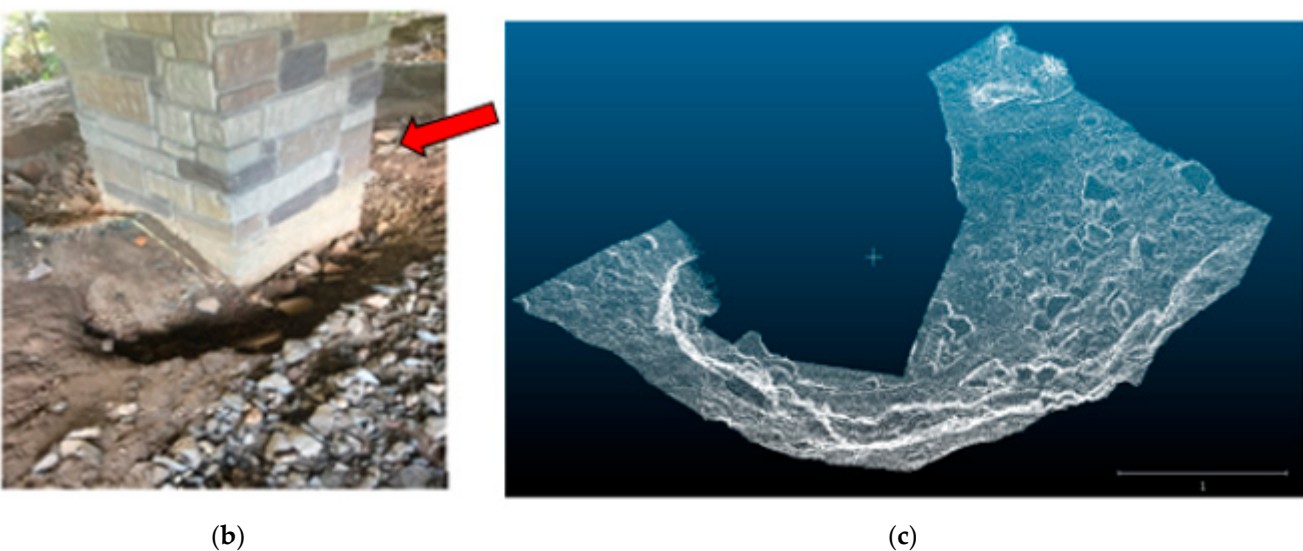

**(b)** **(c)**

**Figure 5.** Scour scan using LiDAR: (**a**) LiDAR scan of scour point cloud (not a river cross-section)—1.5 m; (**b**) image of scour hole (Pier 1); (**c**) scour hole point clouds looking from the direction of arrow in (**b**) (photo credit: S.E. Chen).

## 2. Three-Dimensional Finite Element Modelling

To study the effect of scouring on important structural design parameters, a three-dimensional, non-linear finite element analysis of a single drilled pier foundation supporting the pier (Figure 4b) was developed using the FEA software Abaqus™ by Dassault systems® (Abaqus Version 6.13 Documentation; Karlsson & Sorenson, Inc., Pawtucket, RI, USA, 2013) [33].

The bridge pier of the study site is supported on a 1.68 m diameter, 12.95 m long drilled shaft or drilled pile foundation. The pile protruded 15.0 cm above the ground level and was driven through four-layered heterogenous cohesive-frictional (c-ø soil), as shown in Figure 6. However, the soil is modeled 2.13 m below the tip of the pile in Abaqus™. As suggested by a few other studies including Chen et al. [34]; Karthigeyan et al. [35]; Strömblad [26]. The soil domain is extended to an extent of 10 times the pile (10D) diameter from the centerline to avoid the artificial boundary effect on pile-soil behavior. Thus, the overall dimension of the model assembly in Abaqus™ is 34.34 m × 34.34 m × 12.95 m.

The concrete used for pile construction was an AASHTO [36] class A concrete with characteristic strength ($f_c$) of 31 MPa, Young's modulus ($E_c$) of $2.7 \times 10^4$ MPa, and a poison's ratio of 0.2. Main reinforcement of the pile comprised of structural steel with a yield strength ($f_y$) of 413 MPa, a modulus of elasticity ($E_s$) of $20 \times 10^4$ MPa, and a poisson's ration ($\mu_s$) of 0.2. Furthermore, #4 plain or deformed bars were used as lateral ties. The pile reinforcement was made of twenty-seven vertical #10 rebar with a clear cover of 127 mm, and a hoop reinforcement of #4 rebar with a pitch of 127 mm. In this study, the pile was modeled using linear elastic material properties of concrete and steel as stated above.

The Phillips Road study site comprises different layers of alluvial and residual soil deposits, as shown in Figure 6. Thus, the drilled pier was inserted through multilayered soil deposits of varying thicknesses and finally rested on a weathered rock at its bottom. To describe the complex nature of soil and large deformation arising from stiffness reduction due to scouring, soil substrate was modelled as elastic-plastic nonlinear model. There are different material models available in Abaqus™ that can be used to model the pile-soil interaction. However, in this paper, the most commonly used Mohr–Coulomb plasticity model [25,30] was used to depict the nonlinear behavior of soil. The Mohr–Coulomb yield criteria assumes that a yield function is governed by the maximum shear stress that depends on the normal stress. MC criteria states that

$$\tau = C - \sigma \tan \varphi \tag{1}$$

where $\tau$ = shear stress, $C$ = cohesion intercept of the soil, $\sigma$ = normal stress (negative in compression) and $\varphi$ = angle of internal friction. Soil elastic properties and plasticity parameters required to define MC model for numerical simulation are listed in Table 1. The soil properties used in this study were the in situ soil properties obtained from the site investigation (geotechnical) report of the study site.

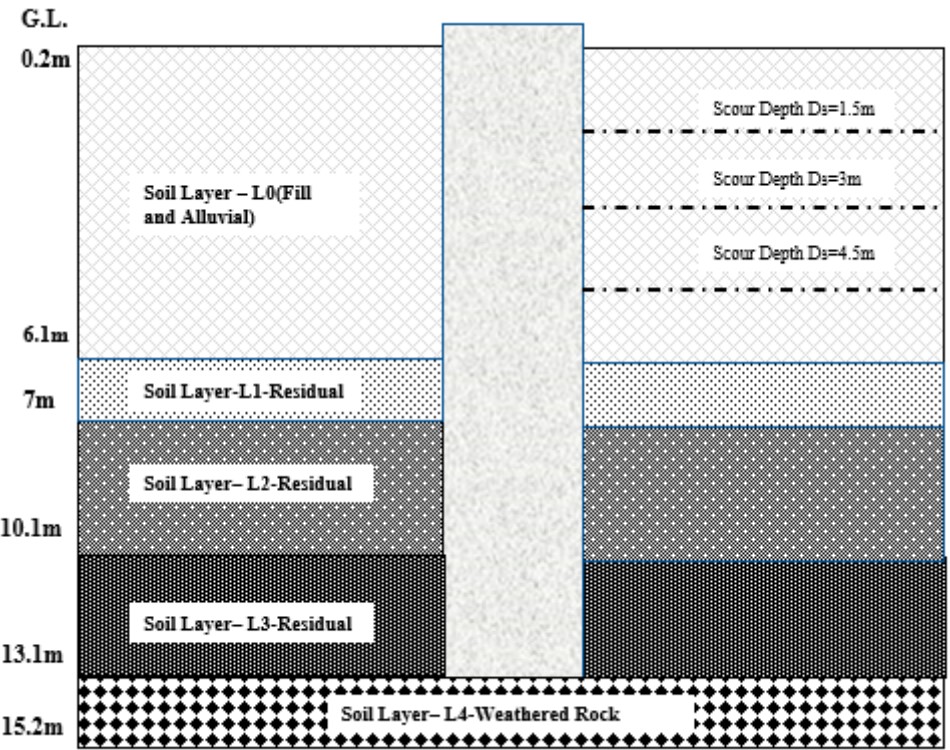

**Figure 6.** Schematic of soil profile and scour depths for the Phillips Road bridge pier.

**Table 1.** Properties of different soil layers at the study site (Phillips Road bridge, UNC Charlotte).

| Soil | Elastic Properties | | | | Mohr–Coulomb Plasticity Parameters | | |
|---|---|---|---|---|---|---|---|
| | Unit Weight ($\gamma s$) (kN/m$^3$) | Young's Modulus (E) (kPa) | Poisson's Ratio ($\upsilon$) | Cohesion Intercept (C) (kPa) | Friction Angle (Ø°) | Dilation Angle ($\psi$°) | Absolute Plastic Strain ($\varepsilon 50$) |
| L0—(c-phi) | 8258 | 47,880 | 0.25 | 1000 | 26 | 0.01 | 0.1 |
| L1—(c-phi) | 9043 | 28,728 | 0.3 | 800 | 26 | 0.01 | 0.01 |
| L2—(c-phi) | 9828 | 76,608 | 0.32 | 4000 | 36 | 0.01 | 0.005 |
| L3—(c-phi) | 12,183 | 95,760 | 0.35 | 72,000 | 40 | 0.01 | 0.00005 |
| L4—(c-phi) | 12,183 | 95,760 | 0.35 | 72,000 | 40 | 0.01 | 0.00005 |

Both pile and soil were modeled as 3D, deformable, solid elements, whereas the longitudinal and transverse reinforcements of the pile were modeled as wire elements [30]. The reinforcement was embedded in a "host region" concrete using "embedded region" interaction property in Abaqus™. Two distinct types of elements were selected for modelling the pile and the soil. Conventional three-dimensional brick elements C3D8 were used to model the soil elements to account for the continuum nature of the soil. The rebar was modeled as a two-node linear 3D truss element T3D2. To minimize the computational time required for analysis, it is typical to model soil close to the pile into a finer mesh and courser mesh for soil more remote from the pile. In this paper, in order to precisely capture the effect of scouring on pile behavior, the soil to be scoured was defined into a very fine

mesh with a pre-defined boundary of a square, while relatively coarser mesh was used for soil more towards the boundaries, as shown in Figure 7. The Abaqus™ model of this study was comprised of a total of 83,345 linear hexahedral elements (C3D8) and 2310 line elements of T3D2 element types.

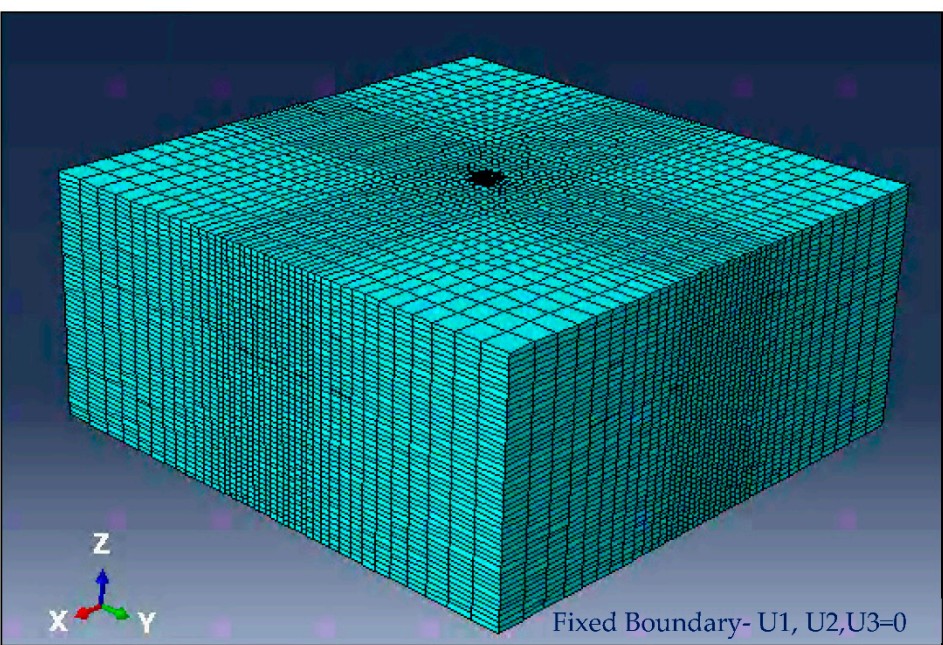

**Figure 7.** Three-dimensional finite element mesh and boundary condition.

### 2.1. Pile-Soil Interaction

The load transfer mechanism of laterally loaded piles depends mainly upon the interaction between the pile and soil. In Abaqus™, the pile-soil interaction was modeled using a small-sliding, surface-to-surface master/slave contact pair formulation [33]. The pile, being stiffer than the surrounding soil, was selected as a master surface, while the soil was selected as a slave surface. The interaction between these two surfaces was defined in terms of normal and tangential behaviors. For depicting the normal behavior between pile and soil, "hard contact" penalty constraint enforcement method was selected. Contact surfaces were allowed to separate after contact with no change in default contact. In tangential direction, penalty friction algorithm [37] along with no limit shear stress parameters were used. Friction coefficient of 0.3 was used to define the friction between the pile and soil contact surfaces. All other settings were kept as default for the analysis. Interaction properties used in the development of the Abaqus FE model are tabulated in Table 2.

The analysis was carried out in the following steps:

Initial → Geostatic → Load Drilled Pier (for no-scour pier)

Initial → Geostatic → Model Change (by Element Removal) → Load Drilled Pier (for scoured pier)

The geostatic step [17] was used to simulate in situ stress conditions in the bridge pier model before applying the design loads on the pier top. User-specified predefined stress field was created by defining the effective vertical stress, $\sigma'$, for each soil layer.

$$\sigma' = \gamma sat \cdot h \tag{2}$$

where $\gamma sat$ is the saturated unit weight of soil and $h$ is the depth to soil layer of interest.

**Table 2.** Interaction properties used in the FE Model.

| | | Properties/Parameters Used in FE Model |
|---|---|---|
| Interaction | Mechanical contact | Surface-to-surface |
| | Sliding formation | Small sliding |
| | Discretization method | Surface-to-surface |
| | Model change (Scour simulation) | Geometry |
| Interaction property | Tangential behavior | - |
| | Friction formulation | Penalty |
| | Friction coefficient | 0.3 |
| | Shear stress | No limit |
| | Normal behavior | |
| | Pressure-overclosure | Hard contact |
| | Constraint enforcement | Penalty |
| | Separation after contact | Allowed |
| | Tie contact | Surface to surface |

The lateral earth pressure coefficient, $k_0$, was then defined to calculate the horizontal stress distribution of the soil. Stresses were calculated for the "geostatic" step, which was in equilibrium with the external loading (gravity load in this case) and boundary conditions, producing zero to negligible deformations.

$$k_0 = 1 - sin\phi \qquad (3)$$

where, $\phi$ = Coefficient of friction for the soil.

### 2.2. Loading and Boundary Conditions

The design vertical load of 2748.86 kN, the design lateral load of 66.72 kN, and the design moment of 149.13 kN-m were calculated as per the AASHTO LRFD method and were applied at the top of the pier through a reference point identified at the top of the pier's cross-section. The degrees of freedom of the elements at the top of the pier were restrained using a kinematic constraint to limit the motion of the coupling nodes to the reference node.

The bottom of the pile was fixed to simulate the embedment of the pile into underlying rock at its tip. Lateral boundaries of the soil surface were restrained against translation in all directions. Figure 7 shows a 3D view of the finite element mesh of the model sans the bridge pier. To simulate scour in the model, soil elements are removed using "Element Removal". We will first discuss the verification of the non-scoured FEM model with the original bridge design.

### 2.3. Verification of Modeling Results

Results of the three-dimensional FEM model were verified by comparing the computed bending moment, and lateral displacement values with those obtained from the L-Pile software used for the pile design of the Phillips Road bridge, which was retrieved from the original geotechnical report. The local scouring was previously not considered in the bridge design and was not considered in the geotechnical investigation. About 6.1 m of soil below ground level is an artificial fill installed after the construction of the foundation, and thus it was not accounted for in the lateral soil resistance calculation of the pier. Accordingly, for verification of the results/comparing the results of Abaqus FE model with L-Pile results, an FE model is developed without considering the top 6.1 m of soil (as shown in Figure 8).

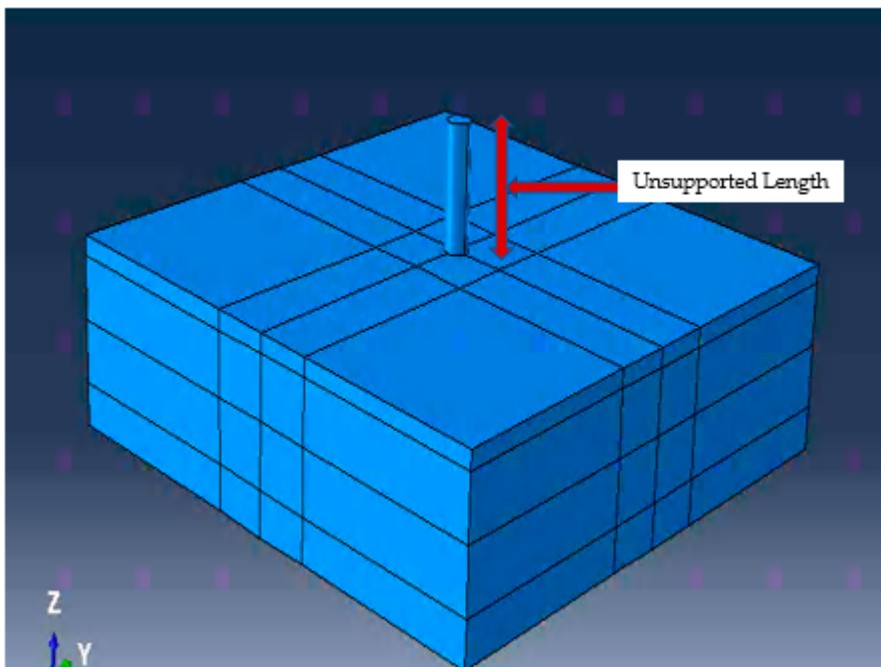

**Figure 8.** Isometric view of the FEM without topsoil layer(fill).

The plot of normalized bending moment values vs. depth from the geotechnical report and the FE models are shown in Figure 9. Both curves show a peak moment at 8 m depth. Comparing the results of the two methods, the p-y method used in the L-Pile program overestimates the bending moment capacity (average = 37.1%). This difference can be attributed to the fundamental/inherent difference between the assumptions made in the two numerical techniques; specifically, L-Pile simplified the geometric effects of the soil into spring elements, whereas the 3D FE modeling is a more realistic simulation of reality.

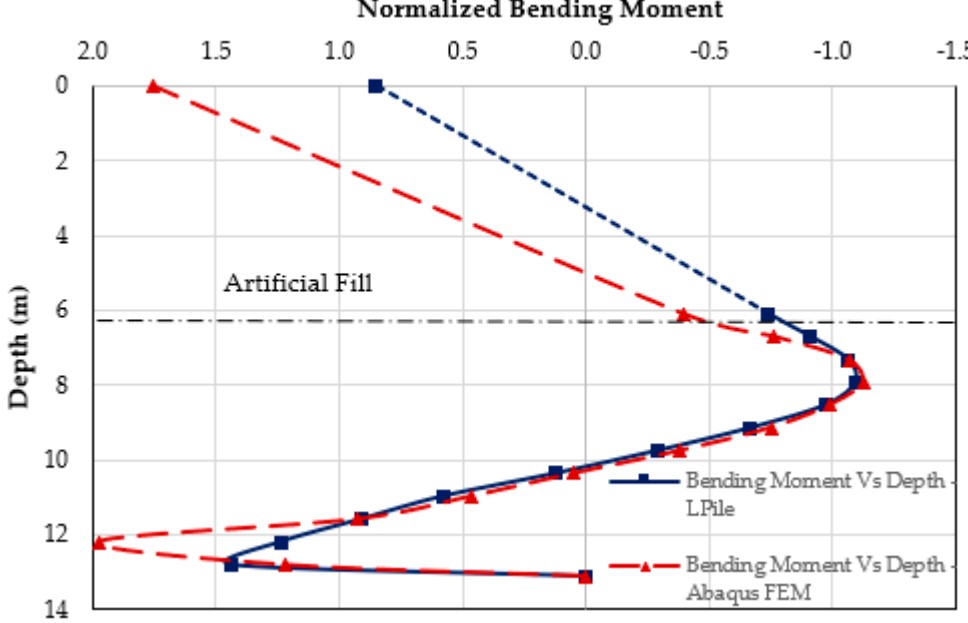

**Figure 9.** Validation of bending moments-L-pile and Abaqus FEM.

Pile-head deflection obtained using the two methods was 2.57 mm and 2.95 mm for the L-Pile (used in geotechnical report) and the FE models, respectively. As the two methods were in good agreement with each other, it could be safely concluded that the assumptions

made and data used to develop the FE model were accurate enough to use the model for investigating scour effects. Table 3 compares the maximum bending moments from L-Pile and FE model results and shows a percentage difference of 43%.

**Table 3.** Model validation data.

| | Validation Model (kN-m) | |
|---|---|---|
| **Depth (m)** | **L-Pile** | **FEM (Without Top 6.1 m Soil, Figure 8)** |
| 6.7 | 600.777 | 339.241 |

*2.4. Scour Hole Dimensions*

Figure 5c shows the local scour hole of 1.5 m diameter formed around a bridge pier. Additionally, as seen in Figure 4b, riverbank erosions (deep cuts) of about 3–3.5 m depth were developed in close proximity to the local scour holes. Considering the intensity of heavy floods in Charlotte, this study assumed/projected a significant widening of local scour holes in the future. As the piers on the northwest side of the case study bridge demonstrated a combination of local and contraction scour, the combined scour problem was worsened due to the accumulation of large quantities of debris in the river channel, which would likely further increase the scour volume in the future.

Thus, this study assumed three different scour hole dimensions for analysis purposes, as shown in Table 4. Despite the uneven and non-symmetric geometry of in situ scour holes, this study used square-shaped scour holes.

**Table 4.** Scour cases.

| Analysis Cases | Scour Hole Dimensions (L × B × D) |
|---|---|
| No Scour | - |
| Case 1: Scour Depth 1.5 m | 1.5 m × 1.5 m × 1.5 m |
| Case 2: Scour Depth 3 m | 3 m × 3 m × 3 m |
| Case 3: Scour Depth 4.5 m | 4.5 m × 4.5 m × 4.5 m |

*2.5. Scour Simulation*

To simulate the scour/erosion of soil surrounding the pier, the region/mesh of a scour hole geometry was removed by defining a distinct analysis step in Abaqus™. "Model Change" interaction in Abaqus™ was used to delete and deactivate the effect of scoured region on the remaining model. This technique is identified as "Element Removal" (ER) and is established as part of the scientific workflow in Abaqus™. As the scoured region described herein is only a result of soil mass loss, scour holes were created by deleting the mesh elements as per the pre-defined square scour hole dimensions. Strömblad [26] modeled scour using a circular shape, which makes modeling a single drilled shaft easier. However, the square shape hole would allow for better modeling of multiple holes or combined scour conditions. Hence, the squared scour hole was used in the current study. Figure 10 shows the square shaped scour region of case 1 (Table 4).

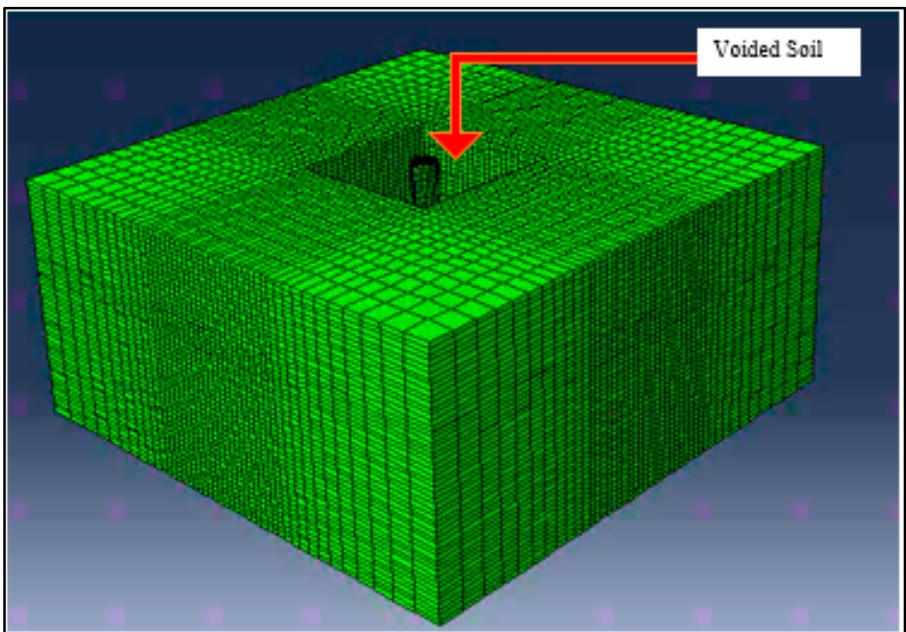

**Figure 10.** Three-dimensional Abaqus model showing scour case 1 (voided area indicates scour hole).

Alternatively, the scour problem could be solved by specifying user defined variables, such as stress state of failure, to automatically remove the elements.

After the ER step, there would not be more interactions between the pile and the scoured region. Forces exerted by the elements/region of the scoured region were ramped down to zero during the ER step; consequently, the effect of the removed scour hole on the rest of the model was completely absent at the end of the ER step [33]. Abaqus performed no further element/stiffness calculations for the removed elements.

For a first-order simulation, the LiDAR detected scour for the bridge pier is depicted as a square hole surrounding the pier. As per the design scour depth mentioned in the geotechnical report, 6.1 m of the soil below ground level (Soil Layer L0) is susceptible to scouring. This level was also in agreement with the scour depths recorded for the last two years 1.4 m to 3.7 m using a terrestrial LiDAR survey. For the case studies considered in this paper, three different scour depths of 1.5 m, 3 m, and 4.5 m (Table 4, Figure 10) were examined to investigate the effect of varying scour levels on bridge pile foundation.

## 3. Results and Discussion

### 3.1. Lateral Displacement vs. Depth Responses

Figure 11 shows the graph of pile deflection computed for three different scour depths viz. 0 m (no scour), 1.5 m, 3 m, and 4.5 m. The figure demonstrates that the scoured cases resulted in significant deflected pier tops with the same design loads. Pile head deflection results are summarized in Table 5. Although the lateral deflection is within the specified limit of maximum allowed pile displacement [36,38], the significantly increased deflection values with the increase in scour draw concerns about potential early instability of the bridge piers. Furthermore, this increase in pile deflection is a result of the increased unsupported length of the pier due to scouring; hence, it is important to recognize the local scour effect on the bridge piers.

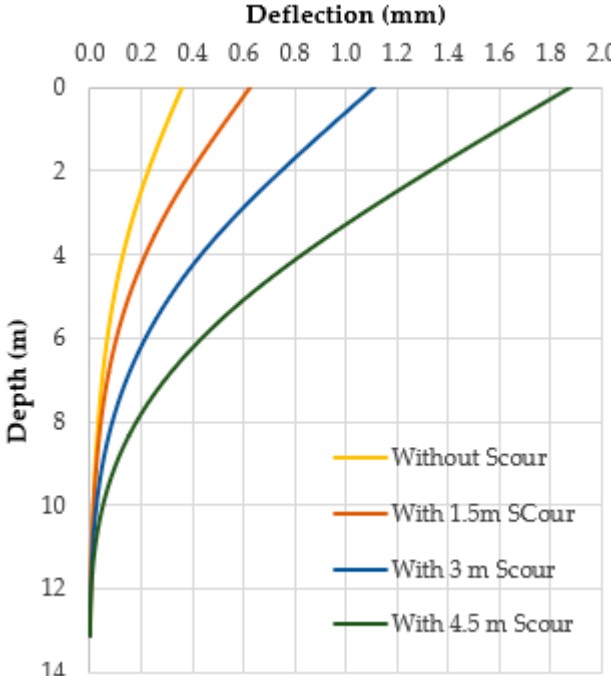

**Figure 11.** Lateral displacement vs. depth for no scour and scour cases 1–3.

**Table 5.** Pile-head deflection (mm) resulted from FEM modeling.

| Scour Case | No Scour | Case 1 (1.5 m) | Case 2 (3 m) | Case 3 (4.5 m) |
|---|---|---|---|---|
| Deflection (mm) | 0.356 | 0.635 | 1.118 | 1.882 |

*3.2. Profiles of the Bending Moment*

Figure 12 shows the computed bending moment curves along the pile depth for no scour case, and three different scour cases described in Table 4. For scour case 1, the maximum bending moment has increased significantly from 45.06 kN-m to 98.64 kN-m for the first 1.5 m of scour depth. The bending moment is further increased by about 62.41% (from 1.5 m to 3 m scour) and 53.55% (3 m to 4.5 m scour depths). The results are summarized in Table 6. Although these bending moments are well within the ultimate bending moment capacities of the pile, a rapid increase in the moment values demonstrated the effect of local scours on the lateral behavior of the pile foundation. For the no scour case, the maximum bending moment occurred at 3 m below the ground level, and at 3.7 m, 4.3 m, and 5.5 m depth, for scour cases 1 to 3, respectively. This shows the increase in the unsupported length of the pile due to scour resulted in increased values of maximum moment. With an increase in scour depth, the lateral resistance of soil, and soil stiffness reduced significantly, resulting in increased lateral loading on the bridge foundation.

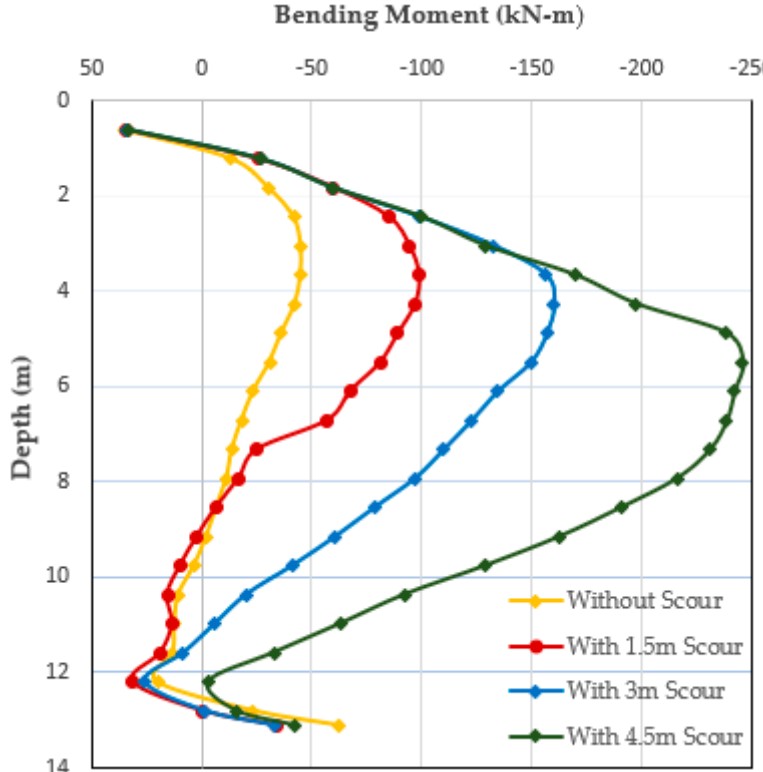

**Figure 12.** Bending moment profiles for no scour and scour cases 1–3.

**Table 6.** Bending moment (kN-m) for different scour scenarios.

| Scour Depth(m) | Max. Bending Moment (kN-m) | Increase with Respect to the No Scour Case (%) |
|---|---|---|
| No Scour | 18.256 | - |
| 1.5 | 56.963 | +212 |
| 3 | 122.961 | +573 |
| 4.5 | 238.734 | +1207 |

Figure 13 shows close-up views of pile displacements at the soil-pile interface, showing evidence of soil-pile separations. Figure 13a shows that the pile displacement before scouring and a modest soil separation exists. Figure 13b shows the pile displacement after scouring and significant separation exists between the pile and surrounding soil. This indicates the nonlinear effects exist in the current model in the form of soil-pile separation, which contributes to the large deformation of the piles after scour. Separation during soil-pile interaction has been described as soil-pile gap formation and can be critical for long-term bridge stability. Gap formation is especially challenging to study [39] and should be considered for future research on the behaviors of piles-on-bank bridges under scour conditions.

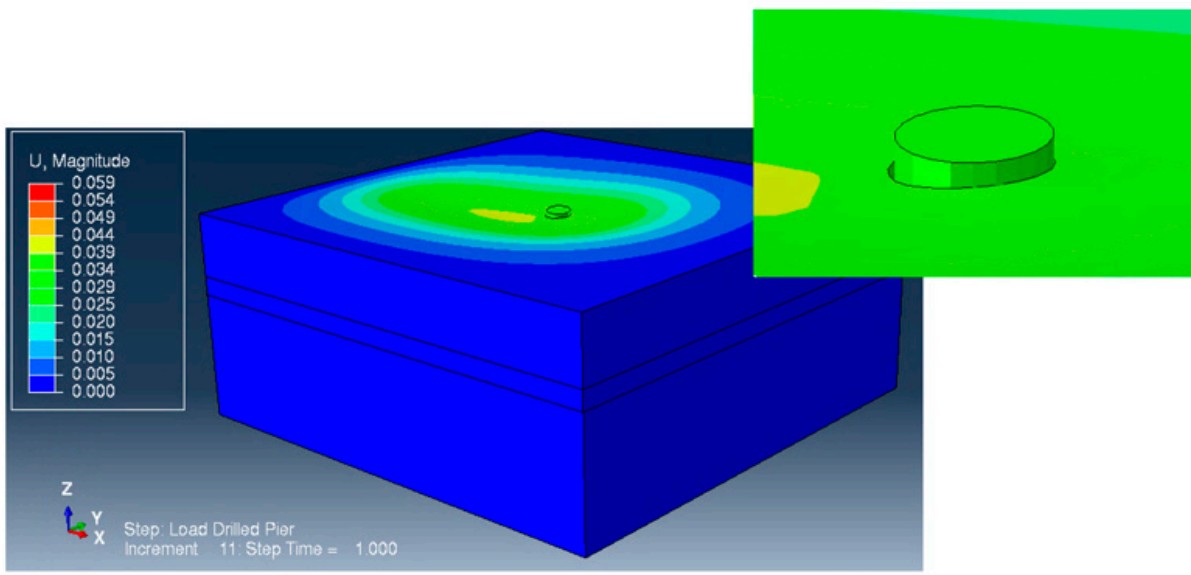

a) Pile displacement (separation from soil – scaled to 500X)

Before scour

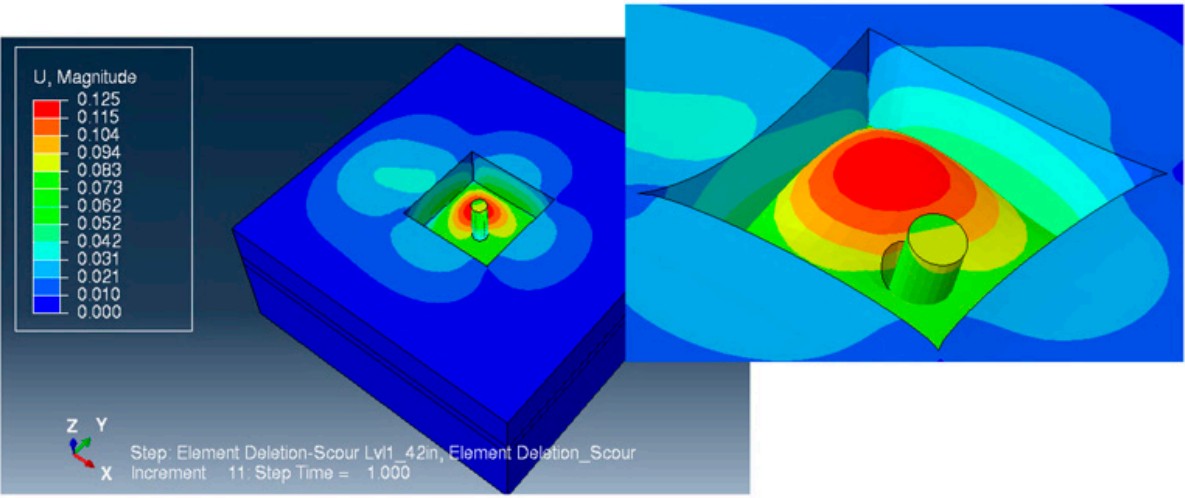

b) Pile displacement (separation from soil – scaled to 500X)

After scour

**Figure 13.** Pile displacements under loading: (**a**) Before scour and (**b**) After scour.

## 4. Conclusions

The Phillips Road bridge above the Toby Creek has its piers on the creek banks (piers-on-bank), and all the piers have experienced local scour problems resulting from the constricted creek cross-section underneath the bridge, as well as the excessive and turbulent flow during several storm flooding events. Due to the steep rise in landscape, a significant number of bridges in eastern North Carolina are bridges with piers-on-bank posing unique local scour problems. To investigate the local scour problem for piers on bank for the Phillips Road bridge, LiDAR scans were performed on the piers and showed that the local scour had diameters of around 1.1 m and 1.5 m for the bridge piers on the east side of the river channel. The observed loss of embankment soil or riverbank was non-uniform along the two bends and could be broadly categorized as contraction scour. The piers on the northwest side of the case study bridge demonstrated a combination of

local and contraction scour, which could further worsen the conditions of the hydraulic system for the bridge due to the accumulation of large quantities of debris in the river channel, which would likely further increase the scour volume in the future.

1.  The scour volume from the LiDAR scans have been used to establish a square scour hole and are applied in the numerical investigation using three-dimensional, nonlinear finite element analysis to study the scouring effect on the structural performance of a bridge pier. Based on the results of simulation, the following conclusions have been drawn: Terrestrial LiDAR scans can be used effectively for frequent and periodic investigations of scour extent. Moreover, the collected site-specific data can be used to establish a possible bridge damage/bridge failure scenario such as estimating the true extents of the scour beyond scour depth estimate.

2.  Based on the results of the 3D non-linear finite element analysis, it can be concluded that the local scour significantly increased the maximum bending moment values in piles, which can increase the possibility of pile failure.

3.  Due to the increase in unsupported length, the lateral deflection of the pile increased considerably with an increase in the scour extent.

4.  A local scour alone can alter noticeably the structural load carrying capacity of the pile as a result of scour. In our study, the effect resulted in a 212% bending moment (see Table 3) for the existing scour at the Phillips Road bridge. This effect would be even worse if local scour was coupled with general scour and/or contraction scour. Thus, bridge foundations need to be frequently inspected and analyzed for the existing scour conditions to take any preventive measures to avoid future damage to the bridge structure.

Finally, scouring between the adjacent piers can significantly reduce the supporting soil mass surrounding the piers on the creek bank and accelerate the deteriorating conditions of the hydraulic structures for the Phillips Road Bridge. Hence, future studies will investigate the effects of multiple piers scouring effects due to interconnection of local scours at each bridge pier.

**Author Contributions:** Conceptualization, V.S.C. and S.-E.C.; methodology, V.S.C., N.B., S.-E.C. and N.S.S.; validation, N.S.S., W.T. and J.D.; formal analysis, V.S.C.; investigation, J.D., C.A., T.C. and T.S.; resources, W.T. and Z.S.; data curation, Z.S.; writing—original draft preparation, V.S.C., S.-E.C. and W.T.; writing—review and editing, J.D., C.A. and N.B.; visualization, N.S.S.; project administration, W.T., S.-E.C., J.D. and C.A.; funding acquisition, W.T., S.-E.C., J.D. and C.A. All authors have read and agreed to the published version of the manuscript.

**Funding:** North Carolina Department of Transportation.

**Institutional Review Board Statement:** Not applicable.

**Informed Consent Statement:** Not applicable.

**Data Availability Statement:** Not applicable.

**Acknowledgments:** The authors would like to acknowledge the technical help from Peter Franz, Campus Architect, University of North Carolina at Charlotte.

**Conflicts of Interest:** The authors declare no conflict of interest.

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
