# Peer review of "Modeling of Progressive Scouring of a Pier-on-Bank"

_2673-4109, doi:10.3390/civileng3020022_

Round 1
Reviewer 1 Report
The authors gather LiDAR profiles around the Phillips Road bridge pier and model the pile below the pier using Abaqus software. Although the scour profile is not particularly of any simple geometric shape, the model around the pier to represent scour is done by removing a "square" region of up to 4.5 m and concluded that local scour significantly increases the maximum bending moment.
The conclusion seems a bit generic but is quantified as up to a 200+% increase in the maximum bending moment.
The actual profile of the LiDAR scan could have been used to get a more realistic estimate of the increase in bending moment, rather than just some "square" removal of elements.
Nevertheless, the case study of the particular bridge considered is interesting, though.
Author Response
Thank you.
Reviewer 2 Report
Over all manuscript presents novel concept. Some physical explanations will improve the quality of the manuscript.
Author Response
Thank you.
Reviewer 3 Report
This paper presents an interesting study on the assessment of the effects of scouring on bridges with piers-on-bank. The limitations of traditional evaluation methods and codes are also shown, compared to more accurate assessment techniques such as finite element modelling with more sophisticated calculation software.
The topic is very interesting, the scientific content of the paper is original and worthy of attention. The paper is well-written and the discussion is satisfactory but repetitive in some of its parts. In this reviewer’s opinion, the paper is ready to be published in the Journal, provided that some minor revisions have to be carried out, according to the following queries:
Figure 1: please put all the pictures at the same size and enlarge them a bit to improve visibility. In addition, it would be interesting to indicate the depicted bridges in the caption.
Figure 4: Please, add a more detailed description of this image in the text, in addition to the caption. Also, specify both in the text and in the caption which pier is represented. At last, it would be better not to split figure and caption in two different pages for better reading.
Page 7, sentence “Figure 5 shows the point cloud of the scour surrounding the selected pier of the Phillips Road bridge from a single scan”: when you recall figure 5, you could describe the content of all pictures in more detail. In particular, picture b has never been explained in the text.
Figure 5: Also in this case, it would be useful to specify which pier is shown, both in the text and in the caption.
Page 9, Sentence “The soil profile of the Phillips Road bridge site is shown in Figure 6”: you could add a more comprehensive comment on the stratigraphy shown in figure 6.
Page 12, Sentence “Results of the three-dimensional FEM model were verified by comparing the computed bending moment, and lateral displacement values with those obtained from L-Pile software used for the pile design of the Phillips Road bridge”: the origin of the comparison data quoted here is not very clear, please add a more detailed description.
Table 3: The table is not clear; it would be better to insert just the "Validation Model (kNm)" portion because it is more relevant at this point of the discussion. In addition, you may explain and comment in the text the results of the bending moments shown in the table. At last, the remaining part of the table "scour simulation model" is more relevant if placed later in section 3.2 where these results are accurately discussed.
Table 4: Since you recall and explain the content of this table in more detail in the next paragraph, you could move the table after paragraph 2.5. In this way, the understanding of the discussion is better.
Figure 10: When you recall the figure in the text, a small explanation of the content shown could be given. In addition, it is written in the caption, but you could also specify in the text which case is depicted.
Table 5: To understand the table at a glance, you could write inside the table that you are summarising the "deflection (mm)" parameter, it is not enough to mention it only in the title.
General comment: Please improve the sharpness of the pictures, most of them appear blurred. At last, correct some typing errors.
